# Genetic Characterization of Carbapenem-Resistant *Klebsiella* spp. from Municipal and Slaughterhouse Wastewater

**DOI:** 10.3390/antibiotics11040435

**Published:** 2022-03-24

**Authors:** Mykhailo Savin, Gabriele Bierbaum, Nico T. Mutters, Ricarda Maria Schmithausen, Judith Kreyenschmidt, Isidro García-Meniño, Silvia Schmoger, Annemarie Käsbohrer, Jens Andre Hammerl

**Affiliations:** 1Institute for Hygiene and Public Health, University Hospital Bonn, 53127 Bonn, Germany; nico.mutters@ukbonn.de; 2Institute of Animal Sciences, University of Bonn, 53115 Bonn, Germany; j.kreyenschmidt@uni-bonn.de; 3Institute for Medical Microbiology, Immunology and Parasitology, Medical Faculty, University of Bonn, 53115 Bonn, Germany; g.bierbaum@uni-bonn.de; 4Department of Hygiene and Environmental Medicine, University Hospital Essen, University of Duisburg-Essen, 45147 Essen, Germany; ricarda.schmithausen@uk-essen.de; 5Department of Fresh Produce Logistics, Hochschule Geisenheim University, 65366 Geisenheim, Germany; 6Department for Biological Safety, German Federal Institute for Risk Assessment, 10589 Berlin, Germany; isidro.garcia-menino@bfr.bund.de (I.G.-M.); silvia.schmoger@bfr.bund.de (S.S.); annemarie.kaesbohrer@bfr.bund.de (A.K.); 7Laboratorio de Referencia de *Escherichia coli* (LREC), Departamento de Microbioloxía e Parasitoloxía, Facultade de Veterinaria, Universidade de Santiago de Compostela (USC), 27002 Lugo, Spain; 8Unit for Veterinary Public Health and Epidemiology, University of Veterinary Medicine, AT-1210 Vienna, Austria

**Keywords:** *Klebsiella pneumoniae*, wastewater, antimicrobial resistance, carbapenem resistance, virulence

## Abstract

Currently, human and veterinary medicine are threatened worldwide by an increasing resistance to carbapenems, particularly present in opportunistic *Enterobacterales* pathogens (e.g., *Klebsiella* spp.). However, there is a lack of comprehensive and comparable data on their occurrence in wastewater, as well as on the phenotypic and genotypic characteristics for various countries including Germany. Thus, this study aims to characterize carbapenem-resistant *Klebsiella* spp. isolated from municipal wastewater treatment plants (mWWTPs) and their receiving water bodies, as well as from wastewater and process waters from poultry and pig slaughterhouses. After isolation using selective media and determination of carbapenem (i.e., ertapenem) resistance using broth microdilution to apply epidemiological breakpoints, the selected isolates (*n* = 30) were subjected to WGS. The vast majority of the isolates (80.0%) originated from the mWWTPs and their receiving water bodies. In addition to ertapenem, *Klebsiella* spp. isolates exhibited resistance to meropenem (40.0%) and imipenem (16.7%), as well as to piperacillin-tazobactam (50.0%) and ceftolozan-tazobactam (50.0%). A high diversity of antibiotic-resistance genes (*n* = 68), in particular those encoding β-lactamases, was revealed. However, with the exception of *bla*_GES-5-like_, no acquired carbapenemase-resistance genes were detected. Virulence factors such as siderophores (e.g., enterobactin) and fimbriae type 1 were present in almost all isolates. A wide genetic diversity was indicated by assigning 66.7% of the isolates to 12 different sequence types (STs), including clinically relevant ones (e.g., ST16, ST252, ST219, ST268, ST307, ST789, ST873, and ST2459). Our study provides information on the occurrence of carbapenem-resistant, ESBL-producing *Klebsiella* spp., which is of clinical importance in wastewater and surface water in Germany. These findings indicate their possible dissemination in the environment and the potential risk of colonization and/or infection of humans, livestock and wildlife associated with exposure to contaminated water sources.

## 1. Introduction

Antimicrobial resistance (AMR) is currently considered one of the major threats to public health and modern healthcare worldwide [1]. In 2015, more than 670,000 infections in the European Union (EU) and European Economic Area (EEA) countries were caused by bacteria resistant to antibiotics, resulting in an estimated 33,000 deaths [2]. Of those, *K. pneumoniae* resistant to third-generation cephalosporins and carbapenems accounted for 84,500 infections in healthcare settings with approx. 6,000 attributable deaths, where this pathogen was most frequently associated with bloodstream and ventilator-associated pneumonia infections [2]. Furthermore, *K. pneumoniae* and *K. oxytoca/K. michiganensis*, which represent the most important clinical species, have been associated with community-acquired infections such as UTIs, meningitis, pneumonia and bacteraemia [3]. In addition, *Klebsiella* spp. are ubiquitous in the environment and have been recovered from surface water, soil, and plants [4].

At 11.3%, *K. pneumoniae* was one of the most commonly reported bacterial species in EU/EEA countries in 2019 among invasive isolates originating from blood or cerebrospinal fluid [5]. In Germany in 2019, less than 1% of clinical *K. pneumoniae* isolates exhibited resistance to carbapenems (i.e., imipenem and/or meropenem); meanwhile, several Southern and East European countries reported rates of more than 10% [6]. However, an increase in carbapenem resistance in *K. pneumoniae* isolates in Germany is clearly noticeable, with a rise from 0.1% in 2015 to 0.9% in 2019 [6]. Furthermore, significantly increasing trends are seen in EU/EEA population-weighted mean percentages of carbapenem resistance among *K. pneumoniae* isolates, from 6.8% in 2015 to 7.9% in 2019 [6].

Currently, the most clinically used carbapenems are meropenem and imipenem [7]. They are the sole, or one of the few, safe and efficacious therapies available for people affected by severe and polymicrobial infections caused by critical priority Gram-negative pathogens, such as multidrug-resistant (MDR) *A. baumannii*, *P. aeruginosa*, and various bacteria of the *Enterobacteriaceae* family [8]. However, amid increasing rates of resistance to carbapenems, e.g., due to the production of carbapenemases, the effectiveness of most β-lactam antimicrobials is compromised. Genes encoding clinically relevant carbapenemases (i.e., KPC, NDM, IMP, OXA-48-like, and VIM), are often located on mobile genetic elements such as plasmids, transposons and intergrons and can be exchanged between *Enterobacteriaceae* and other Gram-negative bacteria, contributing to their spread [9].

While the incidence of carbapenem-resistant *Enterobacteriaceae* (CRE) in the general population is still low (0.3–2.93 infections per 100,000 person-years in USA), they show a high potential to cause outbreaks in healthcare settings [10]. Through clinical wastewater, such high-risk bacterial pathogens are introduced into municipal wastewater systems and are discharged into surface water due to inadequate wastewater treatment. In Germany, a study by Kehl et al. (2021) demonstrated the discharge of a high-risk *K. pneumoniae* clone, ST147, carrying *bla*_NDM_ and *bla*_OXA-48_ from a hospital into surface water [11]. Similar findings of carbapenem-resistant *K. pneumoniae* in rivers with high genetic concordance to clinical isolates have also been reported in other European countries [12,13,14,15].

Carbapenems are restricted to human use only and are not approved for use in veterinary medicine [8,16]. However, the risk of co-resistance to carbapenems, through the use of other antimicrobials in livestock or through horizontal gene transfer from human pathogens, cannot be ruled out [16]. CRE have been sporadically reported in the food chain in various European countries [17]. Carbapenem-resistant, and carbapenemase-producing *K. pneumoniae* have already been detected in poultry, chicken meat, cows and fish in countries that lack strict antimicrobial stewardship in livestock production [18,19,20,21]. However, the transmission of CRE from non-human sources is still limited. Nevertheless, antibiotic resistance remains a notable One Health problem, since not only animals and humans but also the environment is affected by CRE. The aquatic environment is of particular importance, since it provides a basic resource for all ecosystems, including agroecosystems, and holds a crucial role in the dissemination of AMR and their propagation between the natural environment, humans and other animals.

In Germany, the data are still lacking regarding the occurrence of the most clinically relevant species of *Klebsiella* spp. (i.e., *K. pneumoniae* and *K. oxytoca*) with resistance to carbapenems in wastewater, as well as their phenotypic and genotypic characteristics. Thus, the aim of this study is to evaluate the occurrence of carbapenem-resistant *Klebsiella* spp. in municipal wastewater treatment plants (mWWTPs) and their receiving water bodies, as well as in wastewater and process waters from poultry and pig slaughterhouses. In order to better assess their clinical relevance to public and environmental health, we also aim to characterize the recovered *Klebsiella* spp. isolates by applying phenotypic and genotypic methods.

## 2. Results

An overview of the phenotypic antimicrobial resistance of the investigated *Klebsiella* spp. isolates is presented in Figure 1. The isolates exhibit various phenotypic resistances to antimicrobials, inter alia to those highly and critically important for humans (Figure 1). 

As expected, the resistance rates to third- and fourth-generation cephalosporins (i.e., cefotaxime, ceftazidime, cefepime) were high and ranged from 93.3% (28/30) to 100%, whereas the rate of resistance to cefoxitin was lower at 73.3% (22/30). In addition to ertapenem resistance (96.7%, 29/30), as a selection criterion for this study, 16.7% (5/30) and 40.0% (12/30) of the isolates were resistant to imipenem and meropenem, respectively. Notably, 50% (15/30) of the isolates showed resistance to piperacillin-tazobactam and ceftolozan-tazobactam, whereas all of the isolates were susceptible to ceftazidime-avibactam. Almost all of the isolates (96.7%, 29/30) exhibited resistance to fluoroquinolones (i.e., ciprofloxacin), whereas only two isolates (6.7%) were resistant to colistin. Of note, the resistance rates to tigecycline and gentamicin were 20.0% (6/30) and 23.3% (7/30), respectively. The phenotypic resistance patterns of individual isolates are shown in Table 1.

*Klebsiella* spp. isolates represented a reservoir for 68 different ARGs (antimicrobial resistance genes) conferring resistance to antimicrobials belonging to 11 different classes (Table 2).

Of the detected ARGs, 25 encoded β-lactamases which, as expected, were found in all isolates. They encoded enzymes of seven families: *bla*_CTX-M_, *bla*_TEM_, *bla*_SHV_, *bla*_OXA_, *bla*_OXY_, *bla*_GES,_ and *bla*_OKP_. The most abundant were *bla*_CTX-M-15_, *bla*_TEM-1B_ and *bla*_OXY-2-8-like_, accounting for 36.7% (11/30), 30.0% (9/30) and 20.0% (6/30) of the isolates, respectively; and *bla*_OXA-1_, *bla*_SHV-1_ and *bla*_OXA-10_ were each detected in 16.7% (5/30) of the isolates. Of note, combinations of up to four β-lactamases were detected in *K. pneumoniae* isolates from both in- and effluent of mWWTP. Interestingly, *bla*_OXA-1_ and *bla*_OXA-10_, in combination with other extended spectrum β-lactamases of SHV and TEM families, were found to a large extent in isolates with resistance to piperacillin-tazobactam and ceftolozan-tazobactam (Table 1). No carbapenemases were detected, with the exception of *bla*_GES-5-like_ carried by a *K. pneumoniae* isolate recovered from the effluent of mWWTP. Thus, resistance could be mediated by chromosomal alterations.

Among the *Klebsiella* spp. isolates of this study, only some of the genes could be backed to mobile genetic elements. Using the MobileElementFinder tool (version 1.0) from the Center for Genomic Epidemiology, some of the genes could be associated with plasmid or insertion sequences (IS) (Appendix A). Interestingly, IncR plasmids often comprise a combination of the genes *aph*(3″)-Ib*-aph*(6)-Id, leading to a streptomycin resistance phenotype, while some other resistance genes coding for resistances against extended-spectrum β-lactamase antibiotics, tetracycline and sulphonamides were found on other plasmid types (based on the Inc-groups). Overall, the majority of the isolates exhibit a broad diversity of different IS, but mainly IS elements such as ISVsa3, ISAhy2, ISEc9, IS6100, ISKpn19 were found to be associated with resistance determinants. Based on the prevailing data, it cannot be excluded that further genes will be associated with plasmid or IS sequences, due to the use of short-read sequencing data for in silico analysis. Further information about the impact of the plasmids or IS elements on the spread of resistances needs to be determined in detail in another study.

The results of the multilocus sequence-typing (MLST), performed to identify high-risk clones of public health importance, showed that 66.7% (20/30) of the isolates could be assigned to 12 different sequence types (STs). Three isolates each belonged to ST503 and ST2459. ST873, ST458, ST268 and ST252 each accounted for two isolates, whereas the remaining six isolates were identified as ST789, ST441, ST307, ST219, ST1948 and ST16. The STs of seven *K. oxytoca* and three *K. pneumoniae* isolates (Table 1) could not be determined using the prevailing genotyping schemes, possibly indicating new STs.

Among known virulence factors, genes encoding various siderophores, and fimbriae were detected (Table 3). Almost all isolates (96.7%, 29/30) carried genes coding for the enterobactin siderophore system, whereas yersiniabactin, salmochelin, and aerobactin were less prevalent and accounted for 40.0% (12/30) and 33.3% (10/30) of the isolates, respectively. Fimbriae type 1 and fimbriae type 3 were detected in 90.0% (27/30) and 63.3% (19/30) of the isolates, respectively.

Surface polysaccharide locus typing, performed in order to determine the capsule (K antigen) serotypes, showed that the capsule polysaccharide (CPS) types of the vast majority of the isolates (80.0%, 24/30) could be assigned to 14 different types. Three isolates each accounted for KL9 and KL60. KL81, KL74, KL62, KL52, KL21 and KL20 were each represented by two isolates, whereas the remaining six isolates were assigned to KL51, KL24, KL18, KL151, KL114 and KL102.

## 3. Discussion

This study provides novel data on antimicrobial resistance, genetic lineages, virulence factors and CPS-types of carbapenem-resistant (CR) *Klebsiella* spp. from municipal WWTPs as well as process waters and wastewater from German poultry and pig slaughterhouses.

The occurrence of ESBL-producing and CR *Klebsiella* spp. in municipal WWTPs indicates its possible dissemination in the general population and the impact of clinical effluents on the municipal sewer system. Wielders and colleagues (2017) reported an overall prevalence of ESBL-producing *K. pneumoniae* in the general population in the Netherlands of 4.3%, with seasonal differences ranging from 2.6% to 7.4% [22]. Meijs and colleagues (2021) reported even higher levels of ESBL-producing *K. pneumoniae* carriage for veterinary healthcare workers in the Netherlands of 9.8%, emphasizing occupational contact with animals as a potential source of ESBL-producing *K. pneumoniae* in the general population [23]. Several studies have reported high abundances of carbapenemase-producing *Klebsiella* spp. in clinical wastewater and its discharge into the municipal sewer system [11,24,25,26]. The subsequent incidence of such bacteria in surface waters suggests that conventional biological treatment is insufficient in terms of eliminating microbial loads, and shows the negative impact of inadequately treated wastewater on surface waters. Similar findings on ESBL, and on carbapenemase-producing *K. pneumoniae* in Austrian and Swiss rivers mostly within urbanized areas, also highlight the anthropological pollution in aquatic environments [13,14]. *Klebsiella* spp. are known for their ability to survive under adverse conditions and are widely distributed in nature, including in surface water and nutrient rich wastewater [27]. Lepuschitz and colleagues (2019) recovered two multidrug-resistant *K. pneumoniae* ST985 isolates, which share the same cgMLST profile, from sampling sites on a river 200 km apart, demonstrating the possible survival distance of *K. pneumoniae* in river water [13]. A study by Rocha and colleagues (2022) suggests that *Klebsiella* spp. isolates in wastewater retain clinically relevant features, including those acquired through HGT, even after treatment. Thus, further dissemination of the CR isolates recovered in our study among animals and humans, and their colonization and/or infection cannot be ruled out [28]. The application of state-of-the-art wastewater treatment techniques based on oxidative, adsorptive, and membrane-based technologies, as well as the establishment of a surveillance system for clinically relevant antimicrobial-resistant bacteria in surface water should be encouraged.

In this study, almost all isolates exhibited resistance to ciprofloxacin, which is considered to be critically important in human medicine and is often administered to outpatients [29]. Thus, the use of (fluoro)quinolones may contribute to the selection of CR *Klebsiella* spp. in the general community. This finding is in line with the EDCD report indicating that resistance to carbapenems is almost always combined with resistance to other antimicrobial classes, severely narrowing the treatment options for invasive infections caused by “critical pathogens” (i.e., CR *A. baumannii*, *P. aeruginosa* and *Enterobacteriaceae*) and decreasing the likelihood of a positive outcome [6].

Resistance to carbapenems at a clinical level is most frequently caused by the production of carbapenemases, however, other mechanisms may also be involved in the development of such phenotypes [30]. In this study, the results of antimicrobial susceptibility testing were interpreted based on epidemiological cut-off values. This allows the detection of early changes in resistance patterns that could possibly lead to resistance at a clinical level. Nevertheless, no clinically relevant carbapenemases were detected, with only one *K. pneumoniae* isolate from the effluent of mWWTP carrying *bla*_GES-5_. Since no carbapenemases and AmpC β-lactamases were detected, possible mechanisms of resistance to carbapenems, and combinations of β-lactam–β-lactamase inhibitor (i.e., piperacillin-tazobactam and ceftolozane-tazobactam), could be: changes in membrane permeability due to mutations in the genes encoding efflux pump, alterations in the expression and function of porins, and the association of impermeability with the production of ESBL [30]. Furthermore, *bla*_OXA-1_, which encodes a penicillinase with weak affinity for inhibitors such as tazobactam, and hyperproduction of *bla*_TEM-1_ could also be responsible for resistance to piperacillin-tazobactam [31,32]. Narrow-spectrum oxacillinases, e.g., OXA-10-type class D β-lactamases, were previously shown to exhibit weak carbapenemase activity at a level comparable with that of OXA-58 [33,34]. Genes encoding class D β-lactamases are commonly found in *P. aeruginosa*; however, they are also detected in *Enterobacteriaceae*, albeit with lower abundance than in *Pseudomonas* spp., underlying the important role of horizontal gene transfer (HGT) in the spread of AMRs [35]. Resistance to ceftolozane-tazobactam, and ceftazidime-avibactam have been reported in clinical MDR/extensively-drug resistant (XDR) *P. aeruginosa* isolates due to mutations in *bla*_OXA-10_ which developed during antimicrobial treatment [36].

*K. pneumoniae* isolates belonging to the sequence types (STs) determined in this study (ST16, ST252, ST219, ST268, ST307, ST789, ST873, and ST2459) have been detected in various clinical settings across Europe and Asia, causing urinary and respiratory tract infections [37,38,39,40,41,42]. All of them carried different carbapenemases belonging to NDM, OXA, IMP, and KPC families, with some of them exhibiting an extensively drug-resistant (XDR) phenotype. In Europe, the spread of carbapenemases among *K. pneumoniae* is frequently linked to specific clonal lineages such as ST11, ST15, ST101, ST258/512 and their derivatives [43]. However, novel high-risk CR *K. pneumoniae* lineages are continuously emerging. Wyres and colleagues (2018) showed that some *K. pneumoniae* clones are generally better than others at acquiring genetic material via HGT [44]. Currently, comprehensive data on the mechanisms underlying this phenomenon are still lacking, and experimental studies are needed for its further investigation. Interestingly, *K. pneumoniae* ST789 carrying *bla*_NDM-5_ has been reported in neonates in China, and was classified as a novel high-risk CR lineage [45]. In our study *K. pneumoniae* ST789 was detected in wastewater from poultry eviscerators and was carrying *bla*_SHV-25_. However, given the potential of some *Klebsiella* spp. to become high-risk clonal lineage, the possible factors contributing to increased virulence and antimicrobial resistance that might occur in livestock production need to be investigated. This would help develop mitigation strategies in order to interrupt possible dissemination of CRE from livestock to humans. ESBL-producing *Enterobacteriaceae* can serve as a basic model for its spread, since the bacterial species involved are the same and the antimicrobial resistance genes are located on plasmids as well.

The contamination of food of animal and vegetable origin with ESBL-producing *Enterobacteriaceae* is already well described [46,47,48,49,50]. However, considering the risks of CRE to human health, there have been appeals for a zero-tolerance policy and an international ban on the sale of food contaminated with CRE [46]. The fact that carbapenems are not approved for use in veterinary medicine and are predominantly used in human hospital settings can explain the low incidence of carbapenem resistance among the isolates recovered from poultry and pig slaughterhouses. These findings are in line with other reports indicating the absence of or single cases of CRE in European livestock, in particular pigs and broilers [51]. Nevertheless, the risk of AMR transmission through horizontal gene transfer from human pathogens or of co-resistance through the use of other antimicrobials in agriculture cannot be ruled out.

According to epidemiological studies, the first step in the majority of *K. pneumoniae* infections is the colonization of the host’s gastrointestinal tract [52]. The recovered isolates carried genes encoding fimbrial adhesins (type 1 and type 3 fimbriae), which play an essential role in adhesion to the host’s mucosal surfaces and in biofilm formation, as well as genes encoding components of siderophore systems that mediate the uptake of ferric iron [53]. The presence of these virulence factors increases the probability of adherence to the host, colonization, and invasive infections. These data reinforce the recent trend of increasing occurrence of community-acquired *K. pneumoniae* infections in young and healthy individuals, rather than primarily nosocomial infections in immunocompromised patients [54]. However, in comparison to clinical isolates, all of the recovered isolates lacked the factors responsible for the hypermucoviscous phenotype that protects the bacteria against opsonization and phagocytosis.

## 4. Materials and Methods

The sampling sites, procedures and preparation of the samples have been previously described [55,56]. Briefly, the process waters and wastewater (*n* = 87) arising during the operation and cleaning of production facilities were collected from the delivery areas (transport trucks, transport crates, and holding pens) and unclean areas (stunning facilities, scalders, eviscerators, and aggregate wastewater from production facilities) of two poultry and two pig slaughterhouses. In- and effluents (*n* = 62) from their in-house wastewater treatment plants (WWTPs) were also sampled. Further samples (*n* = 36) were taken at two municipal WWTPs (mWWTPs) receiving pretreated wastewater from the pig slaughterhouses, including their on-site preflooders upstream and downstream from the discharge points [55]. At each sampling site, 1 L of water was collected using sterile Nalgene Wide Mouth Environmental Sample Bottles (Thermo Fisher Scientific, Waltham, MA, USA). For further information on selected characteristics of the sampled slaughterhouses, sampling sites and number of samples taken at each sampling site, please see [55,56].

*Klebsiella* spp. isolates with resistance to third-generation cephalosporins, and carbapenems were recovered from water samples by selective cultivation on CHROMagar ESBL and CHROMagar mSuperCarba plates (MAST Diagnostica, Reinfeld, Germany), as previously described [56]. Presumptive colonies of *Klebsiella* spp. were unselectively sub-cultured on Columbia Agar supplemented with 5% sheep blood (*v/v*) (Mast Diagnostics, Reinfeld, Germany). Species identification for the individual isolates was conducted using MALDI-ToF MS (bioMérieux, Marcy-l’Étoile, France) equipped with the Myla software.

Antimicrobial susceptibility testing was performed according to CLSI guidelines (M07-A10), using broth microdilution and applying the epidemiological cut-off values (ECOFFs) from the European Committee on Antimicrobial Susceptibility Testing (EUCAST). In order to assess the clinical relevance of the presumptive ESBL-producing and carbapenem-resistant *Klebsiella* spp. isolates for human medicine, they were tested against the newly approved β-lactam/β-lactamase inhibitor combinations ceftazidime-avibactam, ceftolozan-tazobactam, and piperacillin-tazobactam, by a microdilution method using the clinical cut-off values as previously described [56,57].

A total of 185 *Klebsiella* spp. (155 *K. pneumoniae*, 30 *K. oxytoca*) were isolated, of which 30 (16.2%), comprising 23 *K. pneumoniae* and 7 *K. oxytoca,* showed resistance to at least one of the tested carbapenems (i.e., ertapenem, imipenem, meropenem), and were further investigated in detail. The vast majority (80%, 24/30) originated from mWWTPs (influent, *n* = 12; effluent, *n* = 7) and their on-site preflooders upstream (*n* = 4) and downstream (*n* = 1) from the discharge points. Further isolates were recovered from the process waters and wastewater accruing in poultry (stunning facilities, *n* = 2; eviscerators, *n* = 1) and pig slaughterhouses (pig transporters, *n* = 1; holding pens, *n* = 1; influent in-house chemical-physical WWTP, *n* = 1).

Extraction of genomic DNA (gDNA) from the individual colonies of *Klebsiella* spp., DNA library preparation, and whole-genome sequencing (WGS) were performed as previously described [58]. Briefly, gDNA was extracted using PureLink^®^ Genomic DNA Mini Kit (Invitrogen, Darmstadt, Germany) following the manufacturer’s instructions. Commercial DNA library preparation and WGS were conducted using LGC Genomics GmbH (Berlin, Germany) on an Illumina NextSeq 500/550 V2 (Illumina, CA, USA). De novo assembly of high-quality ~150 bp paired-end sequencing reads was conducted using the SPAdes algorithm of the PATRIC database (v. 3.5.27) [59]. ResFinder v 3.0 and MLST v 2.0 under default values, as well as MyDbFinder (release 1.1; parameters: 90% sequence identity, 60% sequence coverage) of the Center for Genomic Epidemiology (https://cge.cbs.dtu.dk/services/ (accessed on 10 December 2020)) were used for bioinformatics analysis of ARGs, sequence types (STs) and virulence factors, respectively [60]. The captive tool (https://kaptive-web.erc.monash.edu/ (accessed on 10 December 2020)) was used for surface polysaccharide locus typing and variant evaluation. The tool MobileElementFinder (Center for Genomic Epidemiology, version 1.0, default parameters; https://cge.cbs.dtu.dk/services/MobileElementFinder/ (accessed on 10 March 2022) was used for the prediction of mobile genetic elements (MGEs) in combination with plasmid or insertion sequences.

## Figures and Tables

**Figure 1 antibiotics-11-00435-f001:**
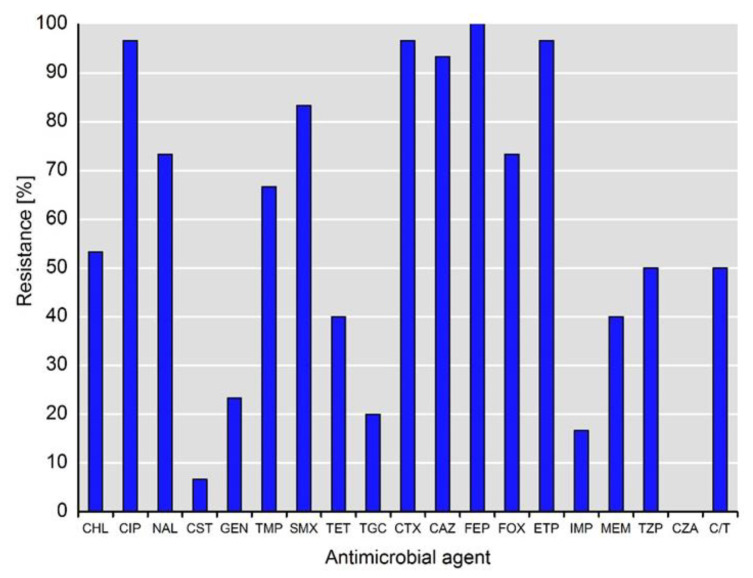
Phenotypical resistance to antimicrobial agents detected among isolates of *Klebsiella* spp. (*n* = 30). Abbreviations for antimicrobial agents: CHL, chloramphenicol; CIP, ciprofloxacin; NAL, nalidixic acid; CST, colistin; GEN, gentamicin; TMP, trimethoprim; SMX, sulfamethoxazole; TET, tetracycline; TGC, tigecycline; CTX, cefotaxime; CAZ, ceftazidime; FEP, cefepime; FOX, cefoxitin; ETP, ertapenem; IMI, imipenem; MEM, meropenem; TZP, piperacillin-tazobactam; CZA, ceftazidime-avibactam; C/T, ceftolozane-tazobactam.

**Table 1 antibiotics-11-00435-t001:** Selected phenotypic and genotypic characteristics of carbapenem-resistant *Klebsiella* spp. isolates recovered from municipal WWTPs and their receiving water bodies as well as from process waters of poultry and pig slaughterhouses.

Isolate	Species	Origin	Resistance Phenotype ^a^	Combinations of β-Lactam–β-Lactamase Inhibitor	Antimicrobial Resistance Genes to Β-Lactams	MLST
05/11-30	*K. oxytoca*	Effluent mWWTP	CIP, NAL, TMP, SMX, CTX, CAZ, FEP, ETP	TZP, C/T	*bla*_OXY-2-8-like_ ^c^	- ^d^
05/11-32	*K. oxytoca*	Effluent mWWTP	CIP, NAL, TMP, SMX, CTX, CAZ, FEP, FOX, ETP	TZP, C/T	*bla* _OXY-2-8-like_	-
05/10-58	*K. oxytoca*	Influent mWWTP	CIP, NAL, TMP, SMX, CTX, CAZ, FEP, FOX, ETP	TZP, C/T	*bla* _OXY-2-8-like_	-
05/10-60	*K. oxytoca*	Influent mWWTP	CIP, NAL, TMP, SMX, CTX, CAZ, FEP, FOX, ETP	TZP, C/T	*bla* _OXY-2-8-like_	-
03/12-04Bki	*K. oxytoca*	On-site preflooder downstream	CIP, NAL, GEN, SMX, CTX, CAZ, FEP, FOX, ETP	TZP, C/T	*bla*_CTX-M-9_, *bla*_OXA-4_, *bla*_OXY-2-8-like_	-
05/13-23	*K. oxytoca*	On-site preflooder upstream	CIP, TMP, SMX, TET, CTX, CAZ, FEP, FOX, ETP	- ^b^	*bla*_CTX-M-15_, *bla*_OXY-2-5_ ^c^	-
05/13-25	*K. oxytoca*	On-site preflooder upstream	CIP, NAL, TMP, SMX, CTX, CAZ, FEP, FOX, ETP	TZP, C/T	*bla* _OXY-2-8-like_	-
03/11-12	*K. pneumoniae*	Effluent mWWTP	CIP, NAL, TET, CTX, CAZ, FEP, FOX, ETP, MEM	-	*bla*_OKP-B-3-like_ ^c^	-
03/11-28	*K. pneumoniae*	Effluent mWWTP	CIP, NAL, TMP, SMX, CTX, CAZ, FEP, ETP	-	*bla*_CTX-M-15_, *bla*_OXA-1_, *bla*_SHV-28_, *bla*_TEM-1B_	ST307
03/11-38	*K. pneumoniae*	Effluent mWWTP	CHL, CIP, NAL, CTX, CAZ, FEP, FOX, ETP, IMI, MEM	-	*bla*_GES-5-like_, *bla*_SHV-2-like_	-
05/11-29	*K. pneumoniae*	Effluent mWWTP	CHL, CIP, NAL, CST, CTX, CAZ, FEP, FOX, ETP, IMI	TZP	*bla*_CTX-M-15_, *bla*_OXA-1_, *bla*_SHV-1_ ^c^, *bla*_SHV-148-like_	ST16
05/11-43	*K. pneumoniae*	Effluent mWWTP	CHL, CIP, NAL, GEN, TMP, SMX, TET, TGC, CTX, CAZ, FEP, FOX, ETP	TZP	*bla*_CTX-M-15_, *bla*_OXY-2-2-like_ ^c^, *bla*_TEM-1B_	-
04/08-35	*K. pneumoniae*	Poultry Eviscerators	CIP, CTX, CAZ, FEP, FOX, ETP	-	*bla* _SHV-25_	ST789
03/06-23	*K. pneumoniae*	Pig Holding Pens	CHL, CIP, TMP, SMX, TET, CTX, CAZ, FEP, ETP	TZP, C/T	*bla*_CTX-M-1_, *bla*_SHV-27-like_, *bla*_TEM-1B_	ST873
03/01-52	*K. pneumoniae*	Influent in-house chemical-physical WWTP	CIP, TMP, SMX, CTX, FEP, FOX, ETP, MEM	-	*bla* _SHV-33_	ST1948
03/10-26	*K. pneumoniae*	Influent mWWTP	CHL, CIP, NAL, TMP, SMX, CTX, CAZ, FEP, FOX, ETP, MEM	TZP, C/T	*bla*_CTX-M-15_, *bla*_OXA-1_, *bla*_SHV-1_	ST2459
03/10-27	*K. pneumoniae*	Influent mWWTP	CHL, CIP, NAL, TMP, SMX, CTX, CAZ, FEP, FOX, ETP, MEM	TZP, C/T	*bla*_CTX-M-15_, *bla*_OXA-1_, *bla*_SHV-1_	ST2459
03/10-46	*K. pneumoniae*	Influent mWWTP	CIP, TMP, SMX, CTX, CAZ, FEP, IMI	-	*bla*_CTX-M-15_, *bla*_SHV-1-like_	ST219
05/10-20	*K. pneumoniae*	Influent mWWTP	CHL, CIP, NAL, GEN, SMX, TET, CTX, CAZ, FEP, FOX, ETP, IMI, MEM	TZP, C/T	*bla*_OXA-10_, *bla*_SHV-31_	ST252
05/10-21	*K. pneumoniae*	Influent mWWTP	CHL, CIP, NAL, GEN, SMX, TET, CTX, CAZ, FEP, FOX, ETP, IMI, MEM	TZP, C/T	*bla*_OXA-10_, *bla*_SHV-31_	ST252
05/10-59	*K. pneumoniae*	Influent mWWTP	CHL, CIP, NAL, GEN, TMP, SMX, TET, TGC, CTX, CAZ, FEP, FOX, ETP	C/T	*bla*_CTX-M-15_, *bla*_SHV-11_, *bla*_TEM-1A_	ST268
05/10-69A	*K. pneumoniae*	Influent mWWTP	CHL, CIP, NAL, SMX, CTX, CAZ, FEP, FOX, ETP, MEM	TZP, C/T	*bla*_OXA-10_, *bla*_SHV-69-like_, *bla*_TEM-1B_	ST503
05/10-69B	*K. pneumoniae*	Influent mWWTP	CHL, CIP, NAL, SMX, FEP, ETP, MEM	TZP, C/T	*bla*_OXA-10_, *bla*_SHV-69-like_, *bla*_TEM-1B_	ST503
05/10-71	*K. pneumoniae*	Influent mWWTP	CHL, CIP, NAL, CST, SMX, CTX, CAZ, FEP, ETP, MEM	TZP, C/T	*bla*_OXA-10_, *bla*_SHV-69-like_	ST503
05/10-83	*K. pneumoniae*	Influent mWWTP	CHL, CIP, NAL, GEN, TMP, SMX, TET, TGC, CTX, CAZ, FEP, FOX, ETP	TZP	*bla*_CTX-M-15_, *bla*_OXA-1_, *bla*_SHV-38-like,_ *bla*_TEM-1B_	ST441
03/13-21	*K. pneumoniae*	On-site preflooder upstream	CHL, CIP, NAL, TMP, CTX, CAZ, FEP, ETP	-	*bla*_CTX-M-15_, *bla*_SHV-1_	ST2459
05/13-31	*K. pneumoniae*	On-site preflooder upstream	CHL, CIP, NAL, GEN, TMP, SMX, TET, TGC, CTX, CAZ, FEP, FOX, ETP	-	*bla*_CTX-M-15_, *bla*_SHV-11_, *bla*_TEM-1A_	ST268
03/05-22	*K. pneumoniae*	Pig Transporters	CHL, TMP, SMX, TET, CTX, CAZ, FEP, ETP	-	*bla*_CTX-M-1_, *bla*_SHV-27-like_, *bla*_TEM-1B_	ST873
01/07-40	*K. pneumoniae*	Poultry Stunning Facilities	CIP, TMP, SMX, TET, TGC, CTX, CAZ, FEP, FOX, ETP, MEM	TZP	*bla*_SHV-28-like_, *bla*_TEM-1B_	ST458
01/07-41	*K. pneumoniae*	Poultry Stunning Facilities	CIP, TMP, SMX, TET, TGC, CTX, CAZ, FEP, FOX, ETP, MEM	-	*bla*_SHV-28-like_, *bla*_TEM-1B_	ST458

^a^ Abbreviations for antimicrobial agents: CHL, chloramphenicol; CIP, ciprofloxacin; NAL, nalidixic acid; CST, colistin; GEN, gentamicin; TMP, trimethoprim; SMX, sulfamethoxazole; TET, tetracycline; TGC, tigecycline; CTX, cefotaxime; CAZ, ceftazidime; FEP, cefepime; FOX, cefoxitin; ETP, ertapenem; IMI, imipenem; MEM, meropenem; TZP, piperacillin-tazobactam; C/T, ceftolozane-tazobactam.; ^b^ susceptible to the combinations of β-lactam–β-lactamase inhibitor TZP, C/T and CZA; ^c^ Intrinsic chromosomally encoded β-lactamases; ^d^ The ST could not be determined using the prevailing scheme.

**Table 2 antibiotics-11-00435-t002:** Prevalence of antibiotic-resistance genes detected in carbapenem-resistant *Klebsiella* spp. isolates recovered from municipal WWTPs and their receiving water bodies as well as from process waters of poultry and pig slaughterhouses.

Antimicrobial Class	Genes	Percentage [%]
β-lactams	*bla* _CTX-M-15_	36.7
	*bla* _TEM-1B_	30.0
	*bla*_OXY-2-8-like_ ^a^	20.0
	*bla* _OXA-1_	16.7
	*bla*_SHV-1_ ^a^	16.7
	*bla* _OXA-10_	16.7
	*bla* _SHV-69-like_	10.0
	*bla*_TEM-1B,_ *bla*_SHV-27-like,_ *bla*_SHV-27-like,_ *bla*_SHV-11,_ *bla*_TEM-1A_	each 6.7
	*bla*_OKP-B-3-like_^a^_,_ *bla*_SHV-28,_ *bla*_GES-5-like,_ *bla*_SHV-2-like,_ *bla*_SHV-148-like,_ *bla*_OXY-2-2-like_^a^_,_ *bla*_SHV-25,_ *bla*_SHV-33,_ *bla*_SHV-38-like,_ *bla*_CTX-M-9,_ *bla*_OXA-4,_ *bla*_OXY-2-5_^a^_,_ *bla*_SHV-28-like_	each 3.3
Aminoglycosides	*strB*	40.0
	*strA*	36.7
	*aadA5*	23.3
	*aac(3)-I*-like	16.7
	*aadA1*	16.7
	*aadA2*	16.7
	*strA*-like	13.3
	*strB*-like	13.3
	*aadB*	10.0
	*aac(3)-IId-like, aacA4, aadA24-like, aph(3′)-Ia*	each 6.7
	*aac(3)-IIa-like, aacA4-like, aadA22, aph(3′)-XV*	each 3.3
Phenicols	*catB3*-like	20.0
	*floR*-like	6.7
	*catB2*	3.3
Fluoroquinolones and aminoglycosides	*aac(6’)Ib-cr*	20.0
	*aac(6’)Ib-cr*-like	10.0
Diaminopyrimidines (Trimethoprim)	*dfrA14*-like	26.7
	*dfrA17*	23.3
	*dfrA1*	13.3
	*dfrA12*	6.7
Sulfonamides	*sul1*	56.7
	*sul2*	33.3
	*sul2*-like	16.7
Phosphonic Acid (Fosfomycin)	*fosA*-like ^a^	56.7
	*fosA* ^a^	13.3
Quinolones	*oqxA*-like ^a^	70.0
	*oqxB*-like ^a^	70.0
	*qnrB66*-like	13.3
	*qnrS1*	13.3
	*qnrA1*-like	10.0
	*qnrB1*	6.7
Tetracyclines	*tet(A)*	13.3
	*tet(A)*-like	6.7
	*tet(B)*	6.7
Macrolides	*mph(A)*	30.0
	*erm(B)*-like	6.7
Lincosamides	*lnu(F)*	3.3

^a^ Intrinsic chromosomally encoded ARGs.

**Table 3 antibiotics-11-00435-t003:** Virulence factors detected in carbapenem-resistant *Klebsiella* spp. isolates recovered from municipal WWTPs and their receiving water bodies as well as from process waters of poultry and pig slaughterhouses.

Virulence Factor	Genes	Percentage ^a^
Enterobactin	*ent*	96.7
Yersiniabactin	*ybt*, *irp1*, *irp2*, *fyuA*	40.0
Salmochelin	*iroN*, *iroBCD*	40.0
Aerobactin	*iucABCD*, *iutA*	33.3
Colibactin	*clbA-R*	0
Regulators of mucoid phenotype	*rmpA*, *rmpA2*, *rmpB*	0
K1 capsule synthesis	*magA*	0
Chromosomal capsule production	*cps*	0
Fimbriae type 1	*fim*	90.0
Fimbriae type 3	*mrk*	63.3

^a^ Percentage of isolates carrying particular virulence factor.

## Data Availability

The data for this study have been uploaded to Sequence Read Archive (SRA). The accession number for the bioproject is PRJNA816413 and the individual biosamples are aslo available there. The temporary number will change as soon as the raw reads are processed by NCBI.

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
