# Peer review of "Genetic Characterization of Carbapenem-Resistant Klebsiella spp. from Municipal and Slaughterhouse Wastewater"

_antibiotics, 2022, doi:10.3390/antibiotics11040435_

Round 1

Reviewer 1 Report

In the manuscript, the Authors present their findings concerning Klebsiella resistance in effluents from water treatment plants and animal handling facilities in Germany. The information gathered is of use to scientific community, and could greatly increase the understanding the origin and spread of antibiotic resistant Enterobacteria.

However, though i have no doubt of the accuracy of data, some parts of the manuscript could be written clearer.

Major remarks

  1. Whole genome sequencing was performed with selected Klebsiella isolates. There is no indication the data was submitted (or will be) to the genome databases.
  2. There should be a clear distinction in the text and analysis between the isolates from wastewater treatment plants/ animal handling facilities and their features - all the isolates are mostly analysed together; comparison in scarce.
  3. The failure to determine MLST in a significant part of isolates is not commented.
  4. Is there any literature data available about Klebsiella from pristine environment? The comparison as already partially mentioned in comment 2 could be very interesting. 
  5. The Results sections would be easier to follow if they would start with short descriptions, explaining the reasons for work performed. 
  6. Colistin resistance and the lack of its ARGs is not commented.
  7. How were the sets of ARGs and virulence genes to be tested/ analysed created? The choice of virulence genes is especially unclear, and is very little commented in discussion section.
  8. Some conclusions are drawn in Discussion section, however section 5 Conclusions contains no text.

Minor remarks

  1. Not all the abbreviations are presented -e.g. WGS, HGT
  2. Table 2 qnr start from capital letters

Reviewer 2 Report

The article is a descriptive study that aims to characterize carbapenem-resistant Klebsiella spp. isolated from wastewater treatment plants, their receiving water bodies and wastewater and process waters from poultry and pig slaughterhouses. The topic is interesting, but there is abundant literature on the subject, including papers by the same authors, and the main conclusion of the study (“their possible dissemination in the environment and potential risk of colonization and/or infection of humans, livestock and wildlife associated with exposure to contaminated water sources”) is also well known. There are also some methodological considerations and the description of the isolates should be more complete.

Minor points

Line 69: Enterobacteriaceae is a family, not a genus.

Table 1: blaOXY-2-5 and blaOXY-2-2-like are also chromosomally encoded betalactamases.

The meaning of "ARG" (antimicrobial resistance genes), abbreviation that is used throughout the manuscript, is mentioned on the last page. It must be specified before use.

Table 2: Chromosomally and plasmidic-encoded genes for beta-lactams resistance are indistinguishable in this table. As presented in table 1, perhaps table 2 should only have determinants of resistance to other antimicrobial families. Are oqxA and B-like located on chromosome and/or plasmids?

Lines 228 and 241: What does HGT mean?

Line 273-275: “However, in comparison to clinical isolates, all recovered isolates lacked factors responsible for production of capsular polysaccharides protecting the bacteria against opsonization and phagocytosis”.  However, in line 168 the authors describe the capsule polysaccharide (CPS) types of the isolates. Perhaps the authors mean that all recovered isolates lacked factors responsible for hypermucoviscous phenotype (rmp, magA)?

I would suggest authors a recent studies which fit very well with research presented (doi: 10.1186/s12864-021-08279-6).

Major points

Lines 296-304: A microorganism is classified as "Resistant" when there is a high likelihood of therapeutic failure even when there is increased exposure, and it is a “Clinical breakpoint”. For a given microbial species and antimicrobial agent, the epidemiological cut-off value (ECOFF) is the highest MIC for organisms devoid of phenotypically-detectable acquired resistance mechanisms. It defines the upper end of the wild-type MIC, and it is a tool for differentiating between wild-type and non-wild type, but the term “resistant” includes other considerations and should not be used in this context. The application of EUCAST values also involves the use of the EUCAST guidelines for broth microdilution. Therefore, if authors have tested antimicrobial susceptibility according to CLSI guidelines, they should apply CLSI clinical breakpoints.

A genome-based phylogenetic analysis should be performed. There are some isolates that appear to be closely related (for example, 05/11-30, 05/11-32, 05/10-58, 05/10-60 and 05/13-25 show the same resistance phenotype and antimicrobial resistance genes to beta-lactams)

No information on mobile genetic elements is included in the manuscript. A brief description about genetic environment of the main resistance-related genes should be included.

The whole genomes shotgun sequences obtained in this study should be deposited in the DDBJ/ENA/GenBank database, and the accession numbers indicated in the manuscript.

Reviewer 3 Report

Comments:  In abstract: ‘12 different sequence types (STs), including clinically relevant ones’. If possible add name of the clinically relevant ST clones.  Line 68, write ‘..and different bacteria of the Enterobacteriaceae family.’  In line 305, please indicate from where the isolates were recovered, from the medium selecting carbapenem-resistant isolates or the ESBL medium?  Line 308, the number was 24 or 25? Since you wrote ‘The vast majority (80%, 24/30) originated from mWWTPs (influent, n=12; effluent, n=7) and their on-site preflooders upstream (n=5 and downstream (n=1) from the discharge points’. (12+9+5+1=25).  In table 3, for example for Yersiniabactin you found ‘ybt, irp1, irp2, fyuA’ and the percentage was 40.0% this percentage is for each of these genes? If yeas please write as in Table 2 ‘each 40..%’ and also for the remaining virulence factors: Salmochelin and Aerobactin and Regulators of mucoid phenotype.  Line 230, write:’.. have already been reported in clinical MDR/extensively-drug resistant (XDR) P. aeruginosa isolates due to mutations……’ Since this is the first time that you wrote XDR.  You can modify the title as follows: Genetic characterization of carbapenem-resistant Klebsiella spp. from municipal and slaughterhouse wastewater in Germany. Or more attractive title.

Round 2

Reviewer 1 Report

The changes the Authors made to manuscript have improved it; to my other remarks the Authors responded clarifying their point. 

Reviewer 2 Report

The manuscript has been
improved